# Glycaemic level and glycaemic variability in acute ischaemic stroke and functional outcome: An observational continuous glucose monitoring study

Lukana Preechasuk[1,2], Siân Rilstone[1], Wen Xi Tang[3], Jackie Man[3], Mingming Yang[3], Erica Zhao[3], Lily Hoque[3], Elif Tuncay[3], Peter Wilding[3], Ian Godsland[1], Omid Halse[3], Soma Banerjee[3], Nick Oliver[1]*, Neil E. Hill[3]

1 Department of Metabolism, Digestion and Reproduction, Imperial College London, South Kensington Campus, London, United Kingdom, 2 Siriraj Diabetes Center of Excellence, Faculty of Medicine Siriraj Hospital, Mahidol University, Bangkok Noi, Bangkok, Thailand, 3 Imperial College Healthcare N.H.S. Trust, St Mary's Hospital, Praed Street, London, United Kingdom

* nickoliver@nhs.net

## Abstract

### Introduction

Glycaemic variability has been associated with poor outcomes in critically ill patients. We aimed to study the association between glycaemic variability and functional outcome in patients with acute ischaemic stroke using continuous glucose monitoring to ensure all episodes of hyper- and hypoglycaemia were captured.

### Research design and methods

Participants with acute ischaemic stroke were enrolled and started blinded continuous glucose monitoring (Dexcom G6) between November 2020 and December 2022. Glucose data from the first 72 hours after admission were analysed. Patients were classified into 3 groups based on change in functional status (Modified Rankin Scale) between admission and discharge. These included (i) remained independent (RI); (ii) deteriorated to dependent (DD); and (iii) remained dependent (RD).

### Results

Data of 67 patients (mean±SD age 72.1±14 years) were analysed; 19 participants had diabetes. The median (IQR) National Institutes of Health Stroke Scale (NIHSS) was 8 (3,14), and 34.3% received reperfusion therapy. The percentage of patients with RI, DD, and RD was 25.4, 55.2, 19.4%. Patients with DD had older age, higher rate of atrial fibrillation, systolic blood pressure, rate of in-hospital infection, NIHSS at admission and at 24 hours after reperfusion therapy compared to those RI. Continuous glucose monitoring was started at 38.4 (29.5,51) hours after stroke onset. Those with DD had higher mean glucose, %time above 180 mg/dL, and glucose standard

**Data availability statement:** Access to the data is restricted by the study ethical approval. As per section 8.3 of the protocol, data will be accessible only to the Research Team and for audit. Applications to collaborate on analyses should be made to the corresponding author. The protocol was approved by Berkshire Research Ethics Committee (REC reference: 20/SC/0214) berkshire.rec@hra.nhs.uk

**Funding:** This work was supported by Dexcom Inc as an Investigator Initiated Study awarded to NH as research funding in his role as Principal Investigator. Dexcom had no role in the study design, data collection and analysis, decision to publish, or preparation of the manuscript.

**Competing interests:** I have read the journal's policy and the authors of this manuscript have the following competing interests: Nick Oliver has received honoraria for speaking and advisory board participation from Abbott Diabetes, Dexcom, Medtronic Diabetes, Tandem Diabetes, Sanofi, and Roche Diabetes, and has received research funding from Medtronic Diabetes and Dexcom. Neil E Hill has received research funding from Dexcom.

**Abbreviations:** AIS, acute ischaemic stroke; CGM, continuous glucose monitoring; CONGA, continuous overlapping net glycaemic action; CV, coefficient of variation; DD, deteriorated to dependent; GV, Glycaemic variability; HASU, Hyper-acute Stroke Unit; IQR, interquartile range; LBGI, low blood glucose index; MAG, mean absolute glucose; MAGE, mean amplitude of glucose excursions; MRS, Modified Rankin Scale; NIHSS, National Institutes of Health Stroke Scale; RD, remained dependent; RI, remained independent; SD, standard deviation; TAR, time above range; TBR, time below range; TIR, time in range; TOAST, the Trial of ORG 10172 in Acute Stroke Treatment.

deviation than the RI group at discharge. Multivariate analysis showed only an association between NIHSS at admission and deterioration in functional status.

## Conclusions

In this pilot study, an association between glycaemic variability and functional deterioration after acute ischaemic stroke was not observed.

## Clinical Trial Registration number

NCT04521634

## Introduction

Stroke is the leading cause of death and disabilities globally. The Global Burden of Diseases, Injuries, and Risk Factors Study (GBD) reported 12·2 million incident cases of stroke and 101 million prevalent cases of stroke in 2019, of which ischaemic stroke constituted 62·4% of all incident strokes [1]. Post stroke hyperglycaemia is common in patients with and without diabetes [2,3] and is associated with poor clinical outcomes including increased infarction size, haemorrhagic transformation, poor functional status and increased mortality [4,5].

The UK Glucose Insulin in Stroke Trial (GIST-UK) [6] and The Stroke Hyperglycaemia Insulin Network Effort (SHINE) [7] randomized clinical trials failed to demonstrate the benefit of intensive glucose control in patients with stroke. Both trials reported higher rate of hypoglycaemic events in the intensively-treated group. The strong evidence of detrimental effects of hyperglycaemia in patients with acute ischaemic stroke, combined with the failure to demonstrate the benefit of intensive glucose management, has drawn attention to another glucose parameter, glucose variability [8].

Glucose variability (GV) [9] is associated with mortality in critically ill patients [10,11]. Previous meta-analysis in patients with acute ischaemic stroke (AIS) using mixed glucose assessment methods also demonstrated that higher acute GV is associated with poor functional outcome within 3 months [12] and early mortality [13]. However, the majority of previous studies used glucose data from point of care testing to calculate GV which fails to capture higher frequency variance between sampling points. There are limited data describing the association between GV from continuous glucose monitoring (CGM), which can provide glucose data every 5 minutes and clinical outcomes in patients with AIS. CGM was utilised to ensure a comprehensive picture of all glycaemic fluctuations including hyper- and hypoglycaemic episodes, which may be missed with fingerprick testing, were captured. Therefore, we aimed to study the impact of GV on functional outcomes in patients with AIS using CGM data.

## Materials and methods

### Study protocol

This was a single centre, prospective observational study. The study protocol was approved by the UK Health Research Authority South Central - Berkshire Research

Ethics Committee (REC reference: 20/SC/0214). The inclusion criteria were adults aged over 18 years admitted with AIS within first 72 hours. The exclusion criteria were patients with haemorrhagic stroke, pregnancy, terminal illness or life expectancy less than one year. Eligible participants were approached by the research team at Hyper-acute Stroke Unit (HASU), Charing Cross Hospital, London between November 2020 and December 2022. Where capacitous, patients provided written informed consent or verbal informed consent with witness signature. When patients could not consent, their relative or a clinician, who was not involved in this study, completed a consultee declaration form if they believed the patient would have no objection to taking part in the study.

After enrolment, a Dexcom G6 CGM sensor (Dexcom, San Diego) was applied to the upper arm or abdomen according with the manufacturers' instruction. A receiver was provided; glucose data were blinded to both participants and health-care professionals. Capillary blood glucose levels were checked to guide treatment in accordance with hospital guidelines.

Glucose data downloaded from receiver after sensor removal. Data within first 72 hours after admission were used to calculate mean glucose, percentage (%) time in range (TIR; 70–180 mg/dL), % time below range (TBR; <70 mg/dL and <54 mg/dL), % time above range (TBR; >180 mg/dL) and GV including standard deviation (SD), % coefficient of variation (CV), mean absolute glucose (MAG), mean amplitude of glucose excursions (MAGE), continuous overlapping net glycaemic action (CONGA) and low blood glucose index (LBGI).

## Data collection

Ischaemic stroke was classified according to the Trial of ORG 10172 in Acute Stroke Treatment (TOAST) classification[14]. Stroke severity was assessed at admission and at 24 hours after receiving reperfusion therapy using National Institutes of Health Stroke Scale (NIHSS) score by neurologists. The score ranges from 0 to 42 where a higher score means a greater neurological deficit. Functional status before admission and at discharge were evaluated using Modified Rankin Scale (MRS). The score ranges from 0 to 6, where score 0 means no symptoms and 6 means dead. Functionally independent was defined as MRS score 0–2, and functionally dependent was defined as MRS score 3–5 [15,16]. We classified functional status change between before admission and at discharge into 3 groups including (i) remained independent (RI); (ii) deteriorated to dependent (DD); and (iii) remained dependent (RD).

Baseline characteristics including age, sex, ethnicity, smoking status, comorbidities, body mass index and blood pressure were recorded at admission. Factors that might affect glucose variability including enteral tube feeding, oral or intravenous steroid usage, antibiotic and glucose lowering treatment were recorded during admission. Readmission and death rate within 3 months after stroke were collected. Recruitment started on 26/11/2020 and finished on 31/12/2022. We had intended to recruit 100 patients with diabetes and 100 without diabetes for this hypothesis-generating study, however the effects of Covid-19 on research staff capacity meant that we were unable to attain this number and opted for a pragmatic time-based recruitment cut-off.

## Statistical analysis

Normally distributed data are presented as mean±standard deviation (SD); non-normally distributed data as median (interquartile range); categorical data as number (percentage). The differences in characteristics between the RI and DD groups were calculated using the unpaired T test, Mann-Whitney U test, and Chi square test, as appropriate. Univariate and multivariate binary logistic regression analysis were used to evaluate the association between predicted factors and functional deterioration (change from functional independent to functional dependent) at discharge. A cutoff of p-value <0.1 from univariate analysis was used to select variables included in multivariate analysis using Enter method. All analysis were conducted using SPSS software version 20.0 (SPSS, Inc., Chicago, IL, USA). A p-value <0.05 was considered statistically significant.

## Results

Seventy-seven participants with AIS were enrolled; 10 patients had no CGM data due to technical errors. Data from 67 participants were analysed. The median duration of monitoring was 27.6 (5.7,48.2) hours. The mean (±SD) age of patients

was 72.1±14 years, and 60% of patients were male, 28% had diabetes, and 81% were functionally independent before admission (Table 1).

Of the participants who were functionally independent before admission, 69% deteriorated to functional dependent (DD) and 31% remained independent (RI). Compared to participants in the RI group, those in the DD group were significantly older, with higher SBP, % atrial fibrillation, stroke severity at admission and after reperfusion therapy, and infection rate. There was no difference in random blood glucose at hospital arrival between the DD and RI groups. CGM data showed the DD group had higher mean glucose, %TAR 180 mg/dl and SD compared to the RI group. There was no oral or intravenous steroid usage during CGM monitoring. Clinical characteristics and glycaemic parameters between groups

**Table 1. Baseline characteristics, stroke characteristic and glycaemic level at admission in people with and without diabetes.**

|  | Total (n=67) | No DM (n=48) | DM (n=19) |
|---|---|---|---|
| Age (years) | 72.1±14 | 70.6±15 | 76.0±9.0 |
| Male gender, n (%) | 40 (59.7) | 29 (60.4) | 11 (57.9) |
| Ethnicity, n (%) | | | |
| - White | 29 (43.3) | 19 (39.6) | 10 (52.6) |
| - Asian | 6 (9.0) | 4 (8.3) | 2 (10.5) |
| - Black | 4 (6.0) | 4 (8.3) | – |
| - Other/not stated | 20 (41.8) | 21 (43.8) | 7 (36.8) |
| Body mass index (kg/m²)ᵃ | 26.9±5.3 | 26.7±5.6 | 27.5±4.0 |
| Systolic blood pressure at arrival (mmHg) | 158±28 | 154±26 | 169±32 |
| Diastolic blood pressure at arrival (mmHg) | 90±18 | 90±18 | 88±18 |
| Smoking, n (%) | 16 (23.9) | 14 (29.2) | 2 (10.5) |
| Comorbidities, n (%) | | | |
| - Hypertension | 40 (59.7) | 23 (47.9) | 17 (89.5) |
| - Dyslipidemia | 35 (52.2) | 18 (37.5) | 17 (89.5) |
| - Congestive heart failure | 8 (11.9) | 4 (8.3) | 4 (21.1) |
| - Atrial fibrillation | 19 (28.4) | 11 (22.9) | 8 (42.1) |
| - Old CVA | 18 (26.9) | 11 (22.9) | 7 (36.8) |
| - Coronary artery disease | 9(13.4) | 5 (10.4) | 4 (21.1) |
| - Peripheral arterial disease | 5 (7.5) | 4 (8.3) | 1 (5.3) |
| Stroke classification, n (%) | | | |
| - Cardio embolism | 29 (43.3) | 22 (45.8) | 7 (36.8) |
| - Large artery atherosclerosis | 26 (38.8) | 16 (33.3) | 10 (52.6) |
| - Small vessel occlusion | 9 (13.4) | 7 (14.6) | 2 (10.5) |
| - Stroke of unknown cause | 3 (4.5) | 3 (6.3) | 0 |
| MRS pre-admission | 1 (0,2) | 0 (0,1.8) | 2 (1,3) |
| Functional independent pre-admission, n (%) | 54 (80.6) | 42 (87.5) | 12 (63.2) |
| NIHSS at admission | 8 (3,14) | 8 (3,15) | 8 (3,10) |
| Blood glucose at admission (mg/dL) | 121 (94,151) | 106 (90,133) | 173 (139,324) |
| HbA1c at admission (%) | (n=38) 6.0 (5.5,7.0) | (n=26) 5.6 (5.4,6.0) | n=12 9.1 (6.7,10.8) |

Data are presented as mean±SD or median (IQR) or n (%).

ᵃdata available in 50 participants. (MRS, Modified Rankin Score; NIHSS, National Institutes of Health Stroke Scale)

Of 67 participants, 23 (34.3%) received reperfusion therapy including thrombolysis (n=9), thrombectomy (n=6), and both thrombolysis and thrombectomy (n=8). The median length of stay on the HASU was 3.6 (2.5,5) days. CGM was started at 38.4 (29.5,51) hours after stroke onset and 26.8 (20.4,36) hours after hospital arrival.

of functional status at discharge are shown in Table 2 (the data for participants who were functionally dependent prior to admission are also included for information). Comparison of the glucose metrics in the cohort without diabetes (n = 42) who either remained independent or deteriorated to dependent showed similar patterns to that of the wider group (with and without diabetes, n = 54).

From univariate logistic analysis for deterioration of functional status, with RI and DD included, variables associated with developing dependency (p-value <0.1) included age, SBP, diabetes, atrial fibrillation, NIHSS score at admission, antibiotic treatment, SD and MAGE. After adjusting for variables, only NIHSS score at admission was associated with deterioration of functional status (Table 3). To avoid collinearity between glycaemic variability measures as indendent variables, two further analyses were undertaken, one with SD and the other with MAGE but the findings remained almost identical with NIHSS significant at p < 0.02 (see footnote Table 3).

Given that the RD group had no 'developed independence' comparison group equivalent to the 'developed dependence' DD group no paired comparisons were made for the RD group, and summary characteristics for the RD group are presented here for information only. In contrast to the RI group, qualitative differences in the RD group included a lower proportion of men, higher SBP, lower prevalence of smoking, lower reperfusion therapy and higher glucose-lowering agents, antibiotic treatment, diabetes, congestive heart failure and atrial fibrillation. Despite these differences, when participants in the RD group were included in the multivariable analysis, NIHSS remained significantly associated with developing dependency (p = 0.01).

We undertook post hoc power calculations, using the data presented in our manuscript as pilot study information, to provide perspective on the strength of our conclusions. Treating each outcome variable as a separate primary outcome, to have detected the differences summarised in Table 2 as significant at p < 0.05 and with 80% power, we would have needed between 80 and 238 participants (for mean glucose concentration and CONGA, respectively, with equal numbers in each group). A fully powered two-group comparison, would have required for 88 participants for SD and 100 participants for MAGE. Powering on the basis of the univariable logistic regression findings (Table 3) gave similar numbers (82 for SD and 104 for MAGE). Those are the minimum numbers needed, minimised by nominating each outcome variable as an alternative primary outcome. If the intention was to evaluate more than one variable in a fully powered study, Bonferroni corrections for multiple testing would be needed, depending on the numbers of outcome variables the study was intended to cover.

## Discussion

This study demonstrated that mean glucose and glucose SD during the 72 hours after hospital admission were higher in patients who deteriorated from independent to dependent functional status at discharge. However, in multivariate analysis only NIHSS at admission was associated with deterioration in functional status.

Ischaemic stroke is caused by an interruption in cerebral blood flow, which leads to reduced delivery of oxygen and glucose to the brain, resulting in disrupted ATP synthesis, energy deficiency, and eventually cell death [17,18]. Dysglycaemia has differing effects on ischaemic brain regions. In core regions, where there is no residual blood flow at all, blood glucose concentration has little effect on brain tissue. In the penumbra region, where there is reduced and also dynamic blood flow, elevated blood glucose can provoke anaerobic metabolism of glucose, lactic acidosis, free radical production and mitochondrial dysfunction [18,19]. Furthermore, in vitro, recurrent hypoglycaemia can increase cell death in response to ischemia and, in vivo, increased infarction size in a rat model of cerebral ischemia [19].

The detrimental effect of GV on target organs might be caused by increased oxidative stress, endothelial dysfunction, inflammation, and hypercoagulability [20]. Few studies have explored the effect of GV on the cerebrovascular system. Glucose fluctuations have been shown to enhance the effect of methylglyoxal, a precursor of advanced glycation end products, on brain microvascular endothelial cell dysfunction [21]. From neuroblastoma cell studies, neuron cells exposed to fluctuating glucose concentrations showed markedly decreased mitochondrial activity compared to constantly high

**Table 2. Clinical characteristics and glycemic parameters between groups of functional status at discharge.**

| | Remained Independent (RI) (n = 17) | Deteriorated to dependent (DD) (n = 37) | Remained dependent (RD) (n = 13) | p-value between RI and DD |
|---|---|---|---|---|
| Age (years) | 62.6 ± 13.5 | 74.4 ± 12.5 | 77.9 ± 11.6 | <0.01 |
| Male gender, n (%) | 11 (64.7) | 25 (67.6) | 4 (30.8) | 0.84 |
| Body mass index (kg/m²)[b] | 27.6 ± 5.9 | 26.6 ± 3.9 | 26.6 ± 7.5 | 0.54 |
| SBP at arrival (mmHg) | 146 ± 15 | 160 ± 30 | 168 ± 31 | 0.03 |
| DBP at arrival (mmHg) | 88 ± 14 | 90 ± 15 | 90 ± 30 | 0.72 |
| Smoking, n (%) | 6 (35.3) | 9 (24.3) | 1 (7.7) | 0.52 |
| MRS pre-admission | 0 (0,1) | 1 (0,1) | 3 (3,3) | 1.00 |
| NIHSS at admission | 5 (2,8) | 9 (4,16.5) | 6 (3,10) | 0.01 |
| Reperfusion therapy, n (%) | 6 (35.3) | 15 (40.5) | 2 (15.4) | 0.71 |
| NIHSS 24 hours after intervention (n = 23) | 1.5 (1,4) | 9 (3,15) | 14.5 | 0.01 |
| Glucose lowering agents[c], n (%) | 1 (5.9) | 8 (21.6) | 4 (30.8) | 0.24 |
| Enteral tube feeding[c], n (%) | 0 | 4 (10.8) | 1 (7.7) | 0.30 |
| Antibiotic treatment[c], n (%) | 1 (5.9) | 12 (32.4) | 4 (30.8) | 0.04 |
| Comorbidities | | | | |
| - Diabetes, n (%) | 1 (5.9) | 11 (29.7) | 7 (53.8) | 0.08 |
| - Hypertension, n (%) | 8 (47.1) | 23 (62.2) | 9 (69.2) | 0.30 |
| - Congestive heart failure, n (%) | 0 | 6 (16.2) | 2 (15.4) | 0.16 |
| - Atrial fibrillation, n (%) | 1 (5.9) | 14 (37.8) | 4 (30.8) | 0.02 |
| - Atherosclerosis, n (%) | 7 (41.2) | 14 (37.8) | 7 (53.8) | 0.82 |
| Glycemic parameters | | | | |
| - Blood glucose at admission, (mg/dL) | 110 (99,140) | 121 (92,166) | 139 (99,189) | 0.75 |
| - HbA1c at admission (%) | 5.8 (5.4,6.2) (n = 8) | 6.0 (5.5,6.8) (n = 24) | 7.4 (5.6,10.9) (n = 6) | 0.40 |
| CGM during first 72 hours after admission | | | | |
| - Mean glucose (mg/dL) | 121 (97,135) | 133 (121,158) | 140 (124,182) | 0.02 |
| - SD | 1.0 (0.9,1.4) | 1.5 (1.0,1.9) | 1.5 (1.1,2.5) | 0.04 |
| - % CV | 17.0 (12.7,20.3) | 18.3 (13.2,23.5) | 16.4 (13.9,22.7) | 0.33 |
| - CONGA | 1.51 (1.17,1.87) | 1.76 (1.29,2.23) | 1.59 (1.37,2.06) | 0.13 |
| - LBGI | 0.25 (0.04,2.12) | 0.28 (0.00,0.35) | 0.02 (0.00,0.31) | 0.11 |
| - MAGE | 2.85 (2.17,3.48) | 4.31 (2.42,4.87) | 3.54 (2.67,5.25) | 0.07 |
| - MAG | 2.61 (2.14,3.74) | 2.47 (2.04,3.05) | 2.56 (1.91,3.08) | 0.41 |
| - % TIR 70–180 mg/dL | 96.7 (91.0,99.7) | 87.8 (75.6,97.5) | 85.7 (50.4,99.2) | 0.12 |
| - % TBR < 70 mg/dL | 0 (0.0,3.0) | 0 (0.0,0.7) | 0 (0.0,0.3) | 0.29 |
| - % TBR < 54 mg/dL | 0 (0.0,1.4) | 0 | 0 | 0.39 |
| - % TAR > 180 mg/dL | 0 (0.0,3.9) | 8.0 (0,15.5) | 11.1 (0.5,41.5) | 0.03 |
| Length of stay in stroke unit, days | 3.1 (2.0,4.7) | 3.4 (2.6,6.2) | 4.1 (2.9,5.0) | 0.19 |
| Re-admission, n (%) | 2 (12.5) | 4 (11.8) | 1 (7.7) | 1.00 |
| Death, n (%) | 0 | 4 (10.8) | 1 (77) | 0.30 |

Data were presented as mean (SD) or median (IQR) or n (%).

[a]p value comparing between RI and DD group.

[b]data available in 50 participants.

[c]during CGM monitoring (SBP, systolic blood pressure; DBP, diastolic blood pressure; CGM, continuous glucose monitoring; MRS, modified Rankin score; NIHSS, National Institutes of Health Stroke Scale; SD, standard deviation; CV, coefficient of variation; CONGA, Continuous overall net glycemic action; LGBI, low blood glucose index; MAGE, mean amplitude of glycaemic excursion; MAG, mean absolute glucose; TIR, time in range; TBR, time below range; TAR, time above range)

**Table 3. Univariate and multivariate logistic regression analysis for functional status deteriorated to dependent at discharge.**

| Variable | Univariate logistic regression analysis | | Multivariate logistic regression analysis[b] | |
|---|---|---|---|---|
| | Odds ratio (95%CI) | p-value | Odds ratio (95%CI) | p-value |
| Age | 1.08 (1.02-1.14) | 0.008 | 1.07 (0.98-1.16) | 0.12 |
| Male gender | 1.14 (0.34-3.81) | 0.84 | | |
| Body mass index[a] | 0.96 (0.83-1.10) | 0.53 | | |
| SBP at arrival | 1.02 (1.00-1.05) | 0.08 | 1.03 (0.98-1.05) | 0.34 |
| DBP at arrival | 1.01 (0.97-1.05) | 0.71 | | |
| Smoking | 0.59 (0.17-2.05) | 0.41 | | |
| NIHSS at admission | 1.16 (1.03-1.30) | 0.01 | 1.21 (1.03-1.42) | 0.02 |
| Reperfusion therapy | 1.25 (0.38-4.12) | 0.71 | | |
| NIHSS 24 hours after intervention | 2.36 (0.80-6.97) | 0.12 | | |
| Antibiotic treatment during CGM monitoring | 7.68 (0.91-64.9) | 0.06 | 2.64 (0.18-38.2) | 0.48 |
| Comorbidities | | | | |
| - Diabetes | 6.77 (0.80-57.5) | 0.08 | 1.93 (0.08-48.6) | 0.69 |
| - Hypertension | 1.85 (0.58-5.90) | 0.30 | | |
| - Congestive heart failure | 3.10 (0.34-28.0) | 0.31 | | |
| - Atrial fibrillation | 9.74 (2.16-81.7) | 0.04 | 1.00 (0.07-14.7) | 1.00 |
| - Atherosclerosis | 0.87 (0.27-2.81) | 0.82 | | |
| Glycemic parameters | | | | |
| - Blood glucose admission | 1.00 (0.99-1.00) | 0.57 | | |
| - HbA1c admission | 1.18 (0.67-2.08) | 0.57 | | |
| CGM during first 72 hours after admission | | | | |
| - Mean glucose | 1.02 (1.00-1.04) | 0.12 | | |
| - SD | 2.85 (0.91-8.96) | 0.07 | 0.22 (0.01-7.09) | 0.40 |
| - % CV | 1.07 (0.97-1.17) | 0.17 | | |
| - CONGA | 2.13 (0.74-6.11) | 0.16 | | |
| - LBGI | 0.78 (0.53-1.15) | 0.22 | | |
| - MAGE | 1.55 (0.96-2.53) | 0.08 | 2.46 (0.58-10.6) | 0.23 |
| - MAG | 0.83 (0.59-1.17) | 0.29 | | |
| - % TIR 70–180 mg/dL | 0.98 (0.96-1.01) | 0.26 | | |
| - % TBR < 70 mg/dL | 0.97 (0.88-1.07) | 0.56 | | |
| - % TBR < 54 mg/dL | 1.00 (0.82-1.23) | 0.98 | | |
| - % TAR > 180 mg/dL | 1.02 (0.99-1.05) | 0.25 | | |

[a]Data available in 50 participants. (SBP, systolic blood pressure; DBP, diastolic blood pressure; CGM, continuous glucose monitoring; NIHSS, National Institutes of Health Stroke Scale; SD, standard deviation; CV, coefficient of variation; CONGA, Continuous overall net glycemic action; LGBI, low blood glucose index; MAGE, mean amplitude of glycaemic excursion; MAG, mean absolute glucose; TIR, time in range; TBR, time below range; TAR, time above range)

[b]SD and MAGE showed marked collinearity (R = 0.93). Therefore, in addition to the analysis with both measures reported here, two further multivariable analyses were carried out, one with SD as GV variable and the other with MAGE. Findings were essentially the same, with NIHSS the only significant independent predictor (both analyses p = 0.02)

glucose [22], while rodent models have demonstrated the association between glucose oscillations and blood-brain barrier dysfunction and altered brain glucose transport [23]. In a rodent model of diabetes, rats with 6 weeks of glucose fluctuation had higher mRNA expression of IL-1β, TNF-α, and abnormal expression of apoptosis-associated genes in the hippocampus [24].

Previous studies that explored the association between GV and stroke outcomes had differences in inclusion criteria, timing of CGM initiation, GV metrics usage, and outcome measurements. In patients with type 2 diabetes who had AIS

within 24 hours NIHSS, HbA1c and MAGE had an association with functional outcome [25]. People with and without diabetes with an acute stroke had an association between early (within 24 hours) glycaemic metrics that reflect hyperglycaemia (including mean glucose, area under the curve >8 mmol/L (144 mg/dL) of blood glucose) and early neurological deterioration and death or dependency at 3 months but there was no association between SD or CV with stroke outcomes [26]. In another study of people with and without diabetes who had acute stroke within 48 hours, higher mean absolute glucose (MAG) was associated with a lower likelihood of neurological improvement during hospitalization [27]. Finally, in patients with AIS involving large vessel anterior circulation occlusion after mechanical thrombectomy SD was associated with in-hospital mortality as well as 3-month mortality [28]. All of these studies recruited differing participants, assessed glucose by different methodologies, and glucose follow-up was both short and variable.

Our study showed higher mean glucose and SD in patients with DD compared to RI group and a trend of association between SD and MAGE with deterioration of functional status at discharge. However, after adjusting for other predictive variables, there was no association between GV and deterioration of functional status. This might be explained by the strong association of stroke severity (NIHSS) and functional outcome and low number of events in the multivariate model compared to adjusting factors. The differing results between our study and others may be explained by the different timing of CGM monitoring after stroke onset and the unselected ischaemic stroke inclusion criteria. Our study started CGM monitoring at 38.4 (29.5,51) hours after stroke onset, mostly limited by the need for magnetic resonance imaging. Therefore, it is possible that early-phase hyperglycaemia and GV more immediately post stroke was missed. A previous CGM study suggested that there was [29] an early-phase of hyperglycaemia during 8 hours after stroke onset, followed by a decrease in glucose at 14–16 hours poststroke, and finally a late-phase hyperglycaemia at 48–88 hours post stroke. Thus, missing early-phase glucose fluctuation in our study might have attenuated the observed relationships between GV and stroke outcomes, and potentially resulting in non-significant results in the multivariate logistic regression analysis.

Previous studies exploring GV in acute stroke have mostly included participants with large vessel occlusion [27,28] or did not classify type of ischaemic stroke [25,26] and some clinical studies have found a non-detrimental effect [30] or even beneficial effect [30,31] of hyperglycaemia in acute lacunar infarction. Our study included 13.4% patients with lacunar infarction whose results could have diluted the effects of GV on the larger dataset. Importantly, our participants experienced minimal, or no, hypoglycaemia, reflecting careful avoidance of exposure to lower glucose values. This may limit our data as previous studies have highlighted that hypoglycaemia may be especially harmful to the injured brain. The lack of association between GV and poor outcome following stroke may reflect the absence of hypoglycaemia in this population rather than a lack of association with GV per se. Our cohort of patients had intrinsically low GV and it is possible that a relationship exists in groups likely to have greater GV (e.g., treated with steroids, insulin and/or sulphonylureas, receiving enteral feeds). Additionally, given the conflicting findings of previous studies, it is possible that GV and stroke outcomes are not intrinsically related and that, when identified, variable GV metrics are biomarkers for other pathophysiological processes or pharmacological interventions caused by or a result of having had a stroke. A larger study would help to discern this.

Participants in a study by Palaiodimou et al, had similar baseline characteristics to our study including CGM initiation's time (32 vs 38.4 hours after stroke onset), NIHSS (9 vs 8), and number of patients without diabetes (81 vs 71%) [27]. They did not find an association between GV and functional independence or excellent functional outcome at 3 months. However, they identified a significant inverse association of greater mean absolute glucose (MAG) and lower likelihood of neurological improvement during hospitalization before and after adjusting for potential confounders. This study included both acute ischaemic and haemorrhagic stroke with younger age of participants (65 vs 72 years) and used anatomical location of stoke rather than stroke severity (NIHSS) to adjust their multivariate analysis. These factors may explain differing results compared to our study.

Challenges with recruitment and a smaller than intended cohort means that the inferences gained from the data should be treated with due caution. Although this was intended as a hypothesis generating study, a greater number of

participants, with more equitable balance between the subgroups, would have made our findings more robust. Our post-hoc power calculations indicate that we would have required 40 people in each group to identify a difference in mean glucose between RI and DD, and 52 people to detect a difference in MAGE, and 44 participants for SD. Even greater numbers might have allowed for greater confidence in the multivariable logistic regression analysis: With DD as the outcome and RI as the reference, there were 7 independent variables in each of separate SD and MAGE analyses but only 54 observations. The so-called 'rule of thumb' for regression analyses is to have at least 10 observations per dependent variable, indicating that at least 70 observations would have been necessary for a more adequate analysis. A further limitation is that we did not record periods when patients were nil-by-mouth; subsequent use of glucose-containing intravenous fluids may have affected their GV. The focus for this hypothesis-generating study was on GV however it may accrue, thus although we did not collect data on the use of intravenous glucose-containing fluids or other interventions, all participants with diabetes had regular fingerprick glucose testing and management of blood glucose was undertaken according in line with local guidelines and clinical assessment by the treating medical team. Other limitations to our study are a lack of glucose data 24 hours after stroke reflecting the need for MRI and delays to assent for participants unable to consent. It was not possible to determine the effects of infection and associated stress hyperglycaemia but it is possible that this contributed to the increased GV seen in the DD group. The study recruited from a single centre in which all participants had the benefit of consultant led multidisciplinary care in a dedicated hyperacute stroke unit, which may limit wider interpretation of the results to other stroke units or hospitals with different practices.

As the use of CGM becomes more widespread in people with type 2 diabetes, and in inpatient settings, challenges we encountered in undertaking this study highlight potentially important clinical implications [32,33]. In particular, the continued use of CGM technology in patients requiring MRI imaging will require additional training and support of both inpatient and imaging staff. The CGM used in this study has been shown to be safe in MRI scanners although national guidelines currently recommend their removal [33,34]. Looking ahead, until this tension is resolved, local guidance and education will be required to enable and implement minimally interrupted use of CGM in people who require MRI scans.

In summary, in an unselected cohort of patients with acute ischaemic stroke, both with and without diabetes, GV metrics were not associated with changes to functional status at discharge when hypoglycaemia was avoided. The use of CGM may enable optimization of glucose and further CGM data closer to an incident acute stroke are still lacking.

## Supporting information

**S1 File. G-VAS Protocol.**
(DOCX)

## Acknowledgments

None

## Author contributions

**Conceptualization:** Siân Rilstone, Omid Halse, Nick Oliver, Neil E. Hill.

**Data curation:** Ian Godsland.

**Formal analysis:** Lukana Preechasuk, Lily Hoque, Elif Tuncay, Neil E. Hill, Ian Godsland.

**Funding acquisition:** Nick Oliver, Neil E. Hill.

**Investigation:** Lukana Preechasuk, Siân Rilstone, Wen Xi Tang, Jackie Man, Mingming Yang, Erica Zhao, Lily Hoque, Elif Tuncay, Peter Wilding, Neil E. Hill.

**Methodology:** Siân Rilstone, Omid Halse, Soma Banerjee, Nick Oliver, Neil E. Hill.

**Supervision:** Siân Rilstone, Peter Wilding, Nick Oliver, Neil E. Hill.

**Writing – original draft:** Lukana Preechasuk, Nick Oliver, Neil E. Hill.

**Writing – review & editing:** Lukana Preechasuk, Wen Xi Tang, Jackie Man, Mingming Yang, Erica Zhao, Lily Hoque, Elif Tuncay, Peter Wilding, Omid Halse, Soma Banerjee, Neil E. Hill.

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
