## [Decision Letter · Decision Letter 0]

13 Sep 2024

PONE-D-24-25866Glycaemic level and glycaemic variability in acute ischaemic stroke and functional outcome at discharge: an observational continuous glucose monitoring studyPLOS ONE

Dear Dr. Oliver,

Thank you for submitting your manuscript to PLOS ONE. After careful consideration, we feel that it has merit but does not fully meet PLOS ONE’s publication criteria as it currently stands. Therefore, we invite you to submit a revised version of the manuscript that addresses the points raised during the review process.

We look forward to receiving your revised manuscript.

Kind regards,

Atakan Orscelik

Academic Editor

PLOS ONE

Journal Requirements:

“I have read the journal's policy and the authors of this manuscript have the following competing interests: Nick Oliver has received honoraria for speaking and advisory board participation from Abbott Diabetes, Dexcom, Medtronic Diabetes, Tandem Diabetes, Sanofi, and Roche Diabetes, and has received research funding from Medtronic Diabetes and Dexcom. Neil E Hill has received research funding from Dexcom.”

Reviewers' comments:

Reviewer's Responses to Questions

**Comments to the Author**

1. Is the manuscript technically sound, and do the data support the conclusions?

Reviewer #1: Partly

Reviewer #2: Yes

Reviewer #3: Yes

2. Has the statistical analysis been performed appropriately and rigorously? 

Reviewer #1: No

Reviewer #2: Yes

Reviewer #3: Yes

3. Have the authors made all data underlying the findings in their manuscript fully available?

Reviewer #1: Yes

Reviewer #2: No

Reviewer #3: Yes

4. Is the manuscript presented in an intelligible fashion and written in standard English?

Reviewer #1: Yes

Reviewer #2: Yes

Reviewer #3: Yes

5. Review Comments to the Author

Reviewer #1: This study investigates the relationship between glycaemic variability and functional outcomes in acute ischaemic stroke patients, finding that stroke severity at admission, rather than glycaemic variability, is the key predictor of functional deterioration at discharge.

While the study offers valuable clinical insights into the relationship between glycaemic variability and functional outcomes in acute ischaemic stroke patients, several limitations should be addressed.

1. The authors initiated glucose monitoring at an average of 38.4 hours after stroke onset, which may have missed short-term glycaemic fluctuations immediately following acute stroke.

2. The sample size of this study needs to be bigger, with only 67 valid cases, which is insufficient to draw convincing conclusions.

3. Diabetes is a risk factor for stroke, and combining glycaemic metrics from stroke patients with diabetes and those without diabetes in the same analysis may be inappropriate.

4. The article does not categorize the types of ischaemic stroke, even though different types may respond differently to hyperglycaemia and glycaemic variability.

5. The glucose monitoring methods may need to be more consistent, as factors such as pre- and post-prandial glucose levels or other interventions like intravenous fluids can significantly affect glucose metrics. Additionally, the use of various medications after stroke, many of which can impact blood glucose levels, was excluded from consideration by the authors.

Reviewer #2: The authors performed a study evaluation the association between glycemic variability and functional outcome. I have the following comments:

1. Methods – please elaborate more on Enter method. Did the authors use the forward or backward or both directions to obtain results from the multivariate analysis?

2. Table 2 – why P-value only for RI vs DD?

3. Did the authors account for other inpatient medical comorbidities? E.g. infection / aspiration pneumonia, a known associated comorbidity after stroke, and their subsequent effect of stress-induced hyperglycemia? E.g. When patient was kept NPO, and thus may have contributed to lower levels of glucose? E.g. when patient had poor oral intake after stroke and was dehydrated so solutions containing glucose e.g. 5% dextrose solution was used to rehydrate the patient?

Reviewer #3: "Glycaemic level and glycaemic variability in acute ischaemic stroke and functional outcome at discharge: an observational continuous glucose monitoring study"

General Overview:

This study aims to explore the relationship between glycaemic variability (GV) and functional outcomes in patients with acute ischaemic stroke (AIS), utilizing continuous glucose monitoring (CGM). The study finds that while GV metrics such as mean glucose and glucose standard deviation (SD) were higher in patients with functional deterioration, only the National Institutes of Health Stroke Scale (NIHSS) at admission was independently associated with worsening functional status in multivariate analysis.

Strengths:

1. Timely Topic: The exploration of GV in the context of stroke is highly relevant, as it contributes to an ongoing discussion about the role of glycaemia in stroke outcomes, especially in light of past conflicting findings on glucose management.

2. Use of Continuous Glucose Monitoring (CGM): The use of CGM provides more granular glucose data than traditional point-of-care tests, adding a novel aspect to the research.

3. Statistical Rigor: The manuscript employs a variety of appropriate statistical techniques, such as logistic regression, and controls for multiple variables in the multivariate analysis.

4. Ethical Approvals and Patient Consent: The study has clear ethical protocols and consent processes, which are meticulously documented.

Major Issues:

1. Study Design and Timing of CGM Initiation:

o Issue: The CGM was initiated at 38.4 hours after stroke onset, potentially missing critical early glucose fluctuations. The study acknowledges this limitation but does not adequately address how it affects the interpretation of GV’s role in the early phases of stroke.

o Recommendation: A deeper discussion about the impact of this delay in CGM initiation on glucose variability (especially hyperglycaemia in the acute phase) would strengthen the manuscript. The authors could explore sensitivity analyses or reference studies where earlier glucose monitoring was performed.

2. Sample Size and Power:

o Issue: The sample size (67 participants) is relatively small, especially given that the study aims to explore associations with multiple variables (NIHSS, glucose variability, comorbidities). Although the authors attribute the limited sample to the impact of COVID-19 on recruitment, this does raise concerns about the study’s power.

o Recommendation: The authors should discuss the implications of this limited sample size more explicitly. A post-hoc power analysis would be beneficial to show whether the sample size was adequate to detect significant associations, particularly for GV metrics like SD and MAGE.

3. Generalizability:

o Issue: The study includes a highly specific population, with 13.4% of patients suffering from lacunar infarctions, which the authors acknowledge might dilute the effect of GV. The broader inclusion criteria (no distinction between stroke subtypes) limit the ability to draw conclusions about certain stroke populations, particularly those with large vessel occlusion.

o Recommendation: The authors should stratify results based on stroke subtype or provide an analysis of how subtype heterogeneity impacts the findings. Subgroup analyses may offer more insights into whether GV plays a more significant role in specific types of strokes.

4. Glycaemic Variability Metrics:

o Issue: While multiple GV metrics were calculated (SD, CV, MAGE, MAG, etc.), the manuscript does not sufficiently discuss why certain metrics might be more meaningful than others in the context of stroke outcomes. For instance, SD is discussed more thoroughly than MAGE, even though the latter is a more direct measure of fluctuations.

o Recommendation: The authors should justify their choice of metrics more robustly. A clearer explanation of why SD and MAGE were prioritized, along with a detailed examination of other GV metrics (CV, CONGA), would be helpful.

5. Interpretation of Multivariate Analysis:

o Issue: The multivariate analysis finds that NIHSS at admission is the only variable significantly associated with functional deterioration. The failure of GV metrics to show significant associations in this analysis is important, but the authors don’t adequately discuss potential reasons for this.

o Recommendation: The manuscript would benefit from a more thorough discussion of why GV, despite being linked to deterioration in univariate analysis, fails to show significance in the multivariate model. Consideration of potential collinearity between NIHSS and glucose levels or other confounders should be discussed.

Minor Issues:

1. Clarity in Statistical Presentation:

o The statistical methods used are sound, but some results could be presented more clearly. For example, confidence intervals for odds ratios in Table 3 should be consistently formatted and displayed to make interpretation easier.

o It would also be helpful to present exact p-values (rather than just stating "<0.05") to provide a better sense of the strength of associations.

2. References to Previous Literature:

o While the authors reference many relevant studies, the discussion could engage more critically with past literature, particularly studies that show conflicting results on the role of GV in stroke outcomes. A more detailed comparison with these studies would contextualize the present findings more effectively.

3. Explanation of Study Limitations:

o The limitations section is adequately addressed but could be expanded. For example, the authors should emphasize the challenges posed by the study’s single-center design, which may limit generalizability to other stroke units or hospitals with different practices.

4. Data Availability Statement:

o The manuscript states that data availability is restricted due to medical confidentiality, which is acceptable. However, the exact procedure for accessing data should be spelled out more clearly to facilitate reproducibility for other researchers.

Conclusion:

This is a well-executed observational study that attempts to shed light on the role of GV in acute ischaemic stroke outcomes. However, limitations such as the timing of CGM initiation and a small sample size make it difficult to draw definitive conclusions. The manuscript could benefit from more robust discussions around the choice of GV metrics, the reasons for the lack of significant associations in multivariate analysis, and how these findings fit into the broader literature. The paper has potential, but further refinement and clarification of these areas are necessary before it can be recommended for publication.

6. PLOS authors have the option to publish the peer review history of their article (what does this mean? ). If published, this will include your full peer review and any attached files.

**Do you want your identity to be public for this peer review?** For information about this choice, including consent withdrawal, please see our Privacy Policy .

Reviewer #1: No

Reviewer #2: No

Reviewer #3: No

---

## [Author Response · Author response to Decision Letter 1]

22 Nov 2024

Reviewer #1: This study investigates the relationship between glycaemic variability and functional outcomes in acute ischaemic stroke patients, finding that stroke severity at admission, rather than glycaemic variability, is the key predictor of functional deterioration at discharge.

While the study offers valuable clinical insights into the relationship between glycaemic variability and functional outcomes in acute ischaemic stroke patients, several limitations should be addressed.

1. The authors initiated glucose monitoring at an average of 38.4 hours after stroke onset, which may have missed short-term glycaemic fluctuations immediately following acute stroke.

Yes, this is a limitation that we struggled to overcome. We had anticipated that we would be able to recruit patients in the Emergency Department and initiate CGM immediately but the reality was, due to a number of factors including but not limited to; challenges seeking consent/assent in the acute setting, clinical expedience, other studies, availability of study personnel to recruit, weekend and overnight admissions, timing of MRI scans, all combined to mean that we failed to start the CGM as promptly as hoped.

2. The sample size of this study needs to be bigger, with only 67 valid cases, which is insufficient to draw convincing conclusions.

Agree – as mentioned above, we struggled with recruitment and were further hampered by the Covid-19 pandemic (page 7, lines 129-131). Whilst 67 is less than we had hoped for it remains a reasonably sized cohort and I hope we have been sufficiently cautious in our interpretation of the data given the number of participants. I’ve reiterated this further in the discussion and highlighted this with the following sentence:

‘Challenges with recruitment and a smaller than intended cohort means that the inferences gained from the data should be treated with due caution. Although this was intended as a hypothesis generating study, a greater number of participants would have made our findings more robust.’

3. Diabetes is a risk factor for stroke, and combining glycaemic metrics from stroke patients with diabetes and those without diabetes in the same analysis may be inappropriate.

We did an analysis of only the people without diabetes, comparing those who remained independent with those who deteriorated to dependent (ie the same analysis as included in Table 2) which I have added below (‘No diabetes’), with the whole cohort’s data (‘Total’) included for comparison. The CGM metrics are similar in the Total group and No diabetes groups; we felt that including the Total group incorporated the richer data. I have included a line in the Results as follows:

‘Comparison of the glucose metrics in the cohort without diabetes (n=42) who either remained independent or deteriorated to dependent showed similar patterns to that of the wider group (with and without diabetes, n=54).’

However, if you would like me to include this table as Supplementary Data we would be happy to do that too.

Total

No diabetes

Remained Independent

(n=17) Deteriorated to dependent

(n=37) p-value Remained Independent

(n=16) Deteriorated to dependent

(n=26) p-value

Age (years) 62.6±13.5 74.4±12.5 0.003 62.5±14 73.6±13.4 0.02

Male gender, n (%) 11 (64.7) 25 (67.6) 0.83 10 (62.5) 18 (69.2) 0.65

Body mass index (kg/m2)b 27.6±5.9 26.6±3.9 0.54 27.6±6 26.2±4 0.46

SBP at arrival (mmHg) 146±15 160±30 0.03 145±15 154±28 0.17

DBP at arrival (mmHg) 88±14 90±15 0.72 88±14 90±15 0.65

Smoking, n (%) 6 (35.3) 9 (24.3) 0.52 6 (37.5) 7 (26.9) 0.51

MRS pre-admission 0 (0,1) 1 (0,1) 1.00 0 (0,1) 0 (0,1) 0.75

NIHSS at admission 5 (2,8) 9 (4,16.5) 0.01 4 (2,8) 10.5 (5,17) 0.004

Reperfusion therapy, n (%) 6 (35.3) 15 (40.5) 0.71 6 (37.5) 10 (38.5) 0.95

NIHSS 24 hours after intervention (n=23) 1.5 (1,4) 9 (3,15) 0.01 1.5 (1,4)

6.5 (3,13) 0.03

Glucose lowering agentsc, n (%) 1 (5.9) 8 (21.6) 0.24 1 (6.3) 1 (3.8) 1.00

Enteral tube feedingc, n (%) 0 4 (10.8) 0.30 0 4 (15.4) 0.28

Antibiotic treatmentc, n (%) 1 (5.9) 12 (32.4) 0.04 0 8 (30.8) 0.02

Comorbidities

-Diabetes, n (%) 1 (5.9) 11 (29.7) 0.08 - - -

-Hypertension, n (%) 8 (47.1) 23 (62.2) 0.30 7 (43.8) 12 (46.2) 0.88

-Congestive heart failure, n (%) - 6 (16.2) 0.16 - 3 (11.5) 0.28

-Atrial fibrillation, n (%) 1 (5.9) 14 (37.8) 0.02 1 (6.3) 8 (30.8) 0.12

-Atherosclerosis 7 (41.2) 14 (37.8) 0.82 6 (37.5) 9 (34.6) 0.85

Glycemic parameters

- Blood glucose at admission, (mg/dL) 110 (99,140) 121 (92,166) 0.75 108 (98,136) 103 (90,132) 0.56

- HbA1c at admission (%) 5.8 (5.4,6.2) 6.0 (5.5,6.8) 0.40 5.6 (5.3,6.0) 5.6 (5.4,6.0) 0.80

CGM data during first 72 hours after admission

Mean glucose (mg/dL) 121 (97,135) 133 (121,158) 0.02 119 (37,133) 125 (119,145) 0.06

SD 1.02 (0.85,1.43) 1.49 (1.00,1.94) 0.04 1.00 (0.82,1.31) 1.31 (0.95,1.65) 0.04

% CV 17.0 (12.7,20.3) 18.3 (13.2,23.5) 0.33 16.9 (12.2,20.3) 18.4 (14.2,23.4) 0.33

CONGA 1.51 (1.17,1.87) 1.76 (1.29,2.23) 0.13 1.51 (1.14,1.73) 1.63 (1.16,2.26) 0.26

LBGI 0.25 (0.04,2.12) 0.28 (0.00,0.35) 0.11 0.53 (0.04,2.12) 0.13 (0.03,0.42) 0.24

MAGE 2.85 (2.17,3.48) 4.31 (2.42,4.87) 0.07 2.76 (2.15,3.45) 3.70 (2.28,4.59) 0.12

MAG 2.61 (2.14,3.74) 2.47 (2.04, 3.05) 0.41 2.76 (2.10,3.91) 2.41 (1.94,2.97) 0.20

% TIR 70-180 mg/dL 96.7 (91.0,99.7) 87.8 (75.6,97.5) 0.12 97.0 (93.2,99.8) 93.9 (83.8,99.7) 0.24

% TBR < 70 mg/dL 0 (0.0,3.0) 0 (0.0,0.7) 0.29 0 (0.0,3.1) 0 (0.0,1.2) 0.32

% TBR < 54 mg/dL 0 (0.0,1.4) 0 0.39 0 (0,1.5) 0 0.29

% TAR > 180 mg/dL 0 (0.0,3.9) 8.0 (0,15.5) 0.03 0 (0,1.9) 4.6 (0,12.3) 0.09

Length of stay in stroke unit, days 3.1 (2.0,4.7) 3.4 (2.6,6.2) 0.19 3.0 (2.0,4.6) 3.4 (2.4,7.1) 0.15

Re-admission, n (%) 2 (12.5) 4 (11.8) 1.00 2 (13.3) 1 (4.0) 0.55

Death, n (%) 0 4 (10.8) 0.30 0 1 (3.8) 1.00

4. The article does not categorize the types of ischaemic stroke, even though different types may respond differently to hyperglycaemia and glycaemic variability. Thank you for raising this. We did classify the types of stroke into cardio-embolic, large artery atherosclerosis, small vessel occlusion and stroke of unknown cause (Table 1) using the Trial of ORG 10172 in Acute Stroke Treatment (TOAST) classification chose not to analyse these groups separately due to the small numbers (and we had not, a priori, planned to).

5. The glucose monitoring methods may need to be more consistent, as factors such as pre- and post-prandial glucose levels or other interventions like intravenous fluids can significantly affect glucose metrics. Additionally, the use of various medications after stroke, many of which can impact blood glucose levels, was excluded from consideration by the authors.

We documented enteral feeding, infection, glucose lowering agents in the different groups (Table 2). Only infection had significant higher number in DD group, which was adjusted in multivariate analysis. Interestingly the (non-significantly) greater use of antibiotics, additional glucose lowering agents and enteral feeding in the DD and RD groups did not correspond with longer duration of time on HASU.

Reviewer #2: The authors performed a study evaluation the association between glycemic variability and functional outcome. I have the following comments:

1. Methods – please elaborate more on Enter method. Did the authors use the forward or backward or both directions to obtain results from the multivariate analysis?

We used the forward method. Only significant data from the univariate analysis was carried forwards for the multivariate analysis.

2. Table 2 – why P-value only for RI vs DD? Sorry, I don’t think we clarified this well in the text. The aim of this analysis was to investigate factors that may have contributed or been associated with functional deterioration (or lack of) hence the statistical analysis was undertaken in patients who were, pre-stroke, independent. The data for people who were dependent prior to their admission on this occasion were included for interest and comparison. I have added a sentence clarifying this:

‘Compared to participants in the RI group, those in the DD group were older, with higher SBP, % atrial fibrillation, stroke severity at admission and after reperfusion therapy, and infection rate’ and ‘Clinical characteristics and glycaemic parameters between groups of functional status at discharge are shown in Table 2 (the data for participants who were functionally dependent prior to admission are also included for information).’

3. Did the authors account for other inpatient medical comorbidities? E.g. infection / aspiration pneumonia, a known associated comorbidity after stroke, and their subsequent effect of stress-induced hyperglycemia? E.g. When patient was kept NPO, and thus may have contributed to lower levels of glucose? E.g. when patient had poor oral intake after stroke and was dehydrated so solutions containing glucose e.g. 5% dextrose solution was used to rehydrate the patient?

We documented enteral feeding, infection, and glucose lowering agents in the different groups (Table 2). Unfortunately, we did not collect information on the duration or number of times when patients were kept nil-by-mouth and this is a limitation of our work. To address this I have added the following sentences in the discussion:

‘A further limitation is that we did not record periods when patients were nil-by-mouth; subsequent use of glucose-containing intravenous fluids may have affected their GV’ and ‘It was not possible to determine the effects of infection and associated stress hyperglycaemia but it is possible that this contributed to the increased GV in the DD group’.

Reviewer #3: "Glycaemic level and glycaemic variability in acute ischaemic stroke and functional outcome at discharge: an observational continuous glucose monitoring study"

General Overview:

This study aims to explore the relationship between glycaemic variability (GV) and functional outcomes in patients with acute ischaemic stroke (AIS), utilizing continuous glucose monitoring (CGM). The study finds that while GV metrics such as mean glucose and glucose standard deviation (SD) were higher in patients with functional deterioration, only the National Institutes of Health Stroke Scale (NIHSS) at admission was independently associated with worsening functional status in multivariate analysis.

Strengths:

1. Timely Topic: The exploration of GV in the context of stroke is highly relevant, as it contributes to an ongoing discussion about the role of glycaemia in stroke outcomes, especially in light of past conflicting findings on glucose management.

2. Use of Continuous Glucose Monitoring (CGM): The use of CGM provides more granular glucose data than traditional point-of-care tests, adding a novel aspect to the research.

3. Statistical Rigor: The manuscript employs a variety of appropriate statistical techniques, such as logistic regression, and controls for multiple variables in the multivariate analysis.

4. Ethical Approvals and Patient Consent: The study has clear ethical protocols and consent processes, which are meticulously documented.

Major Issues:

1. Study Design and Timing of CGM Initiation:

o Issue: The CGM was initiated at 38.4 hours after stroke onset, potentially missing critical early glucose fluctuations. The study acknowledges this limitation but does not adequately address how it affects the interpretation of GV’s role in the early phases of stroke.

o Recommendation: A deeper discussion about the impact of this delay in CGM initiation on glucose variability (especially hyperglycaemia in the acute phase) would strengthen the manuscript. The authors could explore sensitivity analyses or reference studies where earlier glucose monitoring was performed.

Thank you for your comments and suggestions – this has been very helpful for us in attempting to address the issues you have raised.

We have added the following (in red, below the pre-existing text in black) about the impact of delayed CGM initiation, and added further discussion in response to your suggestion to explore the literature more thoroughly in Minor Issues point 2 below.

‘Therefore, it is possible that early-phase hyperglycaemia and GV more immediately post stroke was missed. A previous CGM study suggested that there was 29 an early-phase of hyperglycaemia during 8 hours after stroke onset, followed by a decrease in glucose at 14-16 hours poststroke, and finally a late-phase hyperglycaemia at 48-88 hours post stroke. Thus, missing early-phase glucose fluctuation in our study might have attenuated the observed relationships between GV and stroke outcomes, and potentially resulting in non-significant results in the multivariate logistic regression analysis.’

We also looked at our data again, by tertiles (table below), to see whether the glucose data were affected by start-time of CGM. Only conga was noted to be significantly different amongst the GV metrics.

1st tertile

(0-<31 hours)

(n=22) 2nd tertile

(31-<44 hours)

(n=23) 3rd tertile

(≥ 44 hours)

(n=22) P value

Mean glucose (mg/dL) 128 (117,145) 125 (113,148) 146 (124,187) 0.09

SD 1.25 (0.94, 1.57) 1.32 (0.90, 1.82) 1.21 (1.00, 2.35) 0.47

% CV 18.5 (13.7, 21.3) 18.6 (13.2, 22.3) 16.2 (13.5, 23.7) 0.91

CONGAa 1.45 (1.05, 1.62) 1.80 (1.35, 2.44) 1.82 (1.47, 2.09) 0.03

LBGI 0.19 (0.01, 1.20) 0.16 (0.03, 0.82) 0.03 (0.00, 0.13) 0.06

MAGEb 3.11 (2.04, 3.56) 4.06 (2.83, 4.53) 3.93 (2.44, 5.35) 0.12

MAG 2.31 (1.87, 2.99) 2.79 (1.93, 3.15) 2.56 (2.36, 3.13) 0.25

% TIR 70-180 mg/dL 92.4 (85.7, 98.0) 93.7 (75.6,99.5) 88.5 (47.7, 99.1) 0.68

% TBR < 70 mg/dL 0 (0, 4.9) 0 (0,1.4) 0 0.06

% TBR < 54 mg/dL 0 (0,0.9) 0 0 0.10

% TAR > 180 mg/dL 1.6 (0,11.1) 4.2 (0,15.2) 8.1 (0.5,51.0) 0.29

a. CONGA was calculated from 63 patients

b. MAGE was calculated from 59 patients

2. Sample Size and Power:

o Issue: The sample size (67 participants) is relatively small, especially given that the study aims to explore associations with multiple variables (NIHSS, glucose variability, comorbidities). Although the authors attribute the limited sample to the impact of COVID-19 on recruitment, this does raise concerns about the study’s power.

o Recommendation: The authors should discuss the implications of this limited sample size more explicitly. A post-hoc power analysis would be beneficial to show whether the sample size was adequate to detect significant associations, particularly for GV metrics like SD and MAGE.

We have enlisted the support of a statistician to review the manuscript and ensure any statistical queries are appropriately addressed. We added further discussion to address the issue of low number of events that may have affected the results of multivariate model and added the following sentence:

‘Even though mean glucose and SD were higher in DD group, multivariate analysis did not find the association of glucose level and GV on functional outcome. This might be explained by the strong association of stroke severity (NIHSS) and functional outcome and low number of events in the multivariate model compared to adjusting factors.’

We have, nevertheless, followed the reviewer's suggestion to undertake post hoc power calculations (see table below), using the data presented in our manuscript as pilot study information, which could provide some perspective on the strength of our conclusions. Treating each outcome variable as a separate primary outcome, to have detected the differences summarised in Table 2 as significant at p<0.05 and with 80% power, we would have needed between 80 and 238 participants (for mean glucose concentration and CONGA, respectively, with equal numbers in each group). A fully powered two-group comparison, would have required for SD 88 and MAGE 100. Powering on the basis of the univariable logistic regression findings (Table 3) generally gave similar numbers (82 for SD and 104 for MAGE). Those are, of course, the minimum numbers needed, minimised by nominating each outcome variable as an alternative primary outcome. If the intention was to evaluate more than one variable in a fully powered study, Bonferroni corrections for multiple testing would be needed, depending on the numbers of outcome variables the study was intended to cover. These estimates make it clear that the numbers we recruited fell somewhat sho

---

## [Decision Letter · Decision Letter 1]

4 Dec 2024

PONE-D-24-25866R1Glycaemic level and glycaemic variability in acute ischaemic stroke and functional outcome at discharge: an observational continuous glucose monitoring studyPLOS ONE

Dear Dr. Oliver,

Thank you for submitting your manuscript to PLOS ONE. After careful consideration, we feel that it has merit but does not fully meet PLOS ONE’s publication criteria as it currently stands. Therefore, we invite you to submit a revised version of the manuscript that addresses the points raised during the review process.

We look forward to receiving your revised manuscript.

Kind regards,

Atakan Orscelik

Academic Editor

PLOS ONE

Journal Requirements:

Reviewers' comments:

Reviewer's Responses to Questions

**Comments to the Author**

1. If the authors have adequately addressed your comments raised in a previous round of review and you feel that this manuscript is now acceptable for publication, you may indicate that here to bypass the “Comments to the Author” section, enter your conflict of interest statement in the “Confidential to Editor” section, and submit your "Accept" recommendation.

Reviewer #2: All comments have been addressed

Reviewer #3: (No Response)

2. Is the manuscript technically sound, and do the data support the conclusions?

Reviewer #2: Yes

Reviewer #3: Yes

3. Has the statistical analysis been performed appropriately and rigorously? 

Reviewer #2: Yes

Reviewer #3: No

4. Have the authors made all data underlying the findings in their manuscript fully available?

Reviewer #2: No

Reviewer #3: Yes

5. Is the manuscript presented in an intelligible fashion and written in standard English?

Reviewer #2: Yes

Reviewer #3: Yes

6. Review Comments to the Author

Reviewer #2: (No Response)

Reviewer #3: Title

The title reflects the study design as observational and mentions the key variables. However, it could be more concise without sacrificing clarity. Consider omitting "at discharge" or merging key elements to improve readability.

Abstract

Page 2, Lines 22–42: While the abstract is structured well, the conclusions lack nuance. The assertion that glycaemic variability is not associated with functional deterioration seems too definitive for a pilot study with acknowledged limitations. Suggest softening language to reflect the exploratory nature of findings.

The background section could briefly mention why CGM was chosen over traditional glucose monitoring methods for added context.

Introduction

Page 4, Lines 55–80: The introduction effectively contextualizes the study but needs clarity in explaining why continuous glucose monitoring is uniquely advantageous.

Methods

Variables

Page 6, Lines 104–109: The justification for including specific glycaemic variability metrics is solid. However, the rationale for prioritizing SD and MAG over others in multivariate analysis remains unclear.

Statistical Methods

Page 7, Lines 135–145: The statistical approach is broadly appropriate, but the cutoff for including variables in multivariate analysis (p < 0.1) could lead to overfitting given the small sample size. Provide more details on how collinearity among predictors was assessed.

Results

Participants

Page 8, Lines 147–153: Missing data from 10 participants due to CGM errors should be explored further. Were these missing data random, or could they bias the results?

Descriptive Data

Page 9, Lines 152–157: The division of participants into subgroups (RI, DD, RD) is clear but unbalanced. Consider discussing how this imbalance might impact subgroup analyses.

Outcome Data

Page 10, Lines 173–185: The decision not to include RD in statistical comparisons with RI/DD is reasonable but limits interpretability. A sensitivity analysis incorporating RD could strengthen conclusions.

Main Results

Page 11, Lines 193–198: The multivariate model's failure to show glycaemic variability as significant deserves deeper exploration. Were SD and MAGE highly correlated with NIHSS, thereby diminishing their impact?

Discussion

Key Results

Page 13, Lines 200–208: The emphasis on NIHSS as the primary predictor of functional outcome is appropriate, but the failure of glycaemic variability metrics should be contextualized against prior conflicting findings.

Limitations

The limitation regarding single-center recruitment is mentioned but underemphasized. Additionally, the lack of data on specific interventions (e.g., use of intravenous glucose) during monitoring introduces potential bias.

Interpretation

The authors miss an opportunity to discuss the clinical implications of their findings. How might these results influence the use of CGM in stroke patients or guide future trials?

Supplementary Information

The inclusion of post hoc power calculations is commendable. However, presenting this information graphically could aid reader comprehension.

Minor Issues

Formatting inconsistencies in tables (e.g., Table 2) detract from readability. Ensure consistent decimal places for all variables.

Some abbreviations (e.g., MAG, TIR) are not defined upon first use in the main text.

7. PLOS authors have the option to publish the peer review history of their article (what does this mean? ). If published, this will include your full peer review and any attached files.

**Do you want your identity to be public for this peer review?** For information about this choice, including consent withdrawal, please see our Privacy Policy .

Reviewer #2: No

Reviewer #3: No

---

## [Author Response · Author response to Decision Letter 2]

11 Jan 2025

Reviewer #3:

Title

The title reflects the study design as observational and mentions the key variables. However, it could be more concise without sacrificing clarity. Consider omitting "at discharge" or merging key elements to improve readability.

Modified as suggested

Abstract

Page 2, Lines 22–42: While the abstract is structured well, the conclusions lack nuance. The assertion that glycaemic variability is not associated with functional deterioration seems too definitive for a pilot study with acknowledged limitations. Suggest softening language to reflect the exploratory nature of findings.

Thank you – I’ve done this – changed from ‘Glycaemic variability measured by continuous glucose monitoring started on the second day was not associated with functional deterioration after acute ischaemic stroke’ to ‘In this pilot study, an association between glycaemic variability and functional deterioration after acute ischaemic stroke was not observed’.

The background section could briefly mention why CGM was chosen over traditional glucose monitoring methods for added context.

Added ‘…to ensure all episodes of hyper and hypoglycaemia were captured’.

Introduction

Page 4, Lines 55–80: The introduction effectively contextualizes the study but needs clarity in explaining why continuous glucose monitoring is uniquely advantageous.

Added ‘CGM was utilised to ensure a comprehensive picture of all glycaemic fluctuations including hyper- and hypoglycaemic episodes, which may be missed with fingerprick testing, were captured.’

Methods

Variables

Page 6, Lines 104–109: The justification for including specific glycaemic variability metrics is solid. However, the rationale for prioritizing SD and MAG over others in multivariate analysis remains unclear.

These were the two metrics that were significant in the univariate analysis so carried forwards into the multivariate analysis.

Statistical Methods

Page 7, Lines 135–145: The statistical approach is broadly appropriate, but the cutoff for including variables in multivariate analysis (p < 0.1) could lead to overfitting given the small sample size. Provide more details on how collinearity among predictors was assessed.

There are two related issues here, which, hopefully, we have clarfied in our revised manuscript. To take collinearity first, amomg the 8 variables that related to change from independent to dependent at p<0.1 that were included in our multivariable logistic regression, the two measures of glycaemic variability, SD and MAGE, were very highly correlated with a coefficient of 0.93 and variance inflation factors >7. In contrast, the other 6 variables showed no higher intercorrelation than 0.51 and variance inflation factors no higher than 2. We should, therefore, in the final multivariable logistic analysis have included two separate analyses, one that included only SD as a measure of glycaemic variability and one that included only MAGE. We have now carried out these separate analyses and found that the independent association with NIHSS on admission remains entirely unaffected and no other associations appear as significant. This is now reported as a footnote to Table 3 and in the Results section text (lines 192-195) of our revised manuscript.

The second Issue relates to weakness in the power of our analysis, which was referred to in our original Discussion in relation to the RI and DD between-group comparisons. We have now extended that section (lines 325-329) with a mention of the number of variables included in the multivariable analysis in relation to the number of observations. This further confirms that our study must be regarded as only providing pilot-level information and an indication of the greater numbers of observations required.

Results

Participants

Page 8, Lines 147–153: Missing data from 10 participants due to CGM errors should be explored further. Were these missing data random, or could they bias the results?

The missing data were random – just when the CGM device did not connect – no likelihood of bias

Descriptive Data

Page 9, Lines 152–157: The division of participants into subgroups (RI, DD, RD) is clear but unbalanced. Consider discussing how this imbalance might impact subgroup analyses.

We agree that the subgroups are imbalanced and whilst we have attempted to adjust for this in the analyses do accept that it may impact our results and hence conclusions. I have strengthened the discussion to emphasis this with the following statement:

‘Although this was intended as a hypothesis generating study, a greater number of participants, with more equitable balance between the subgroups, would have made our findings more robust.’

Outcome Data

Page 10, Lines 173–185: The decision not to include RD in statistical comparisons with RI/DD is reasonable but limits interpretability. A sensitivity analysis incorporating RD could strengthen conclusions.

The clinical interest in the data we present here must be in the transitions detected, the better, hopefully, to identify factors that might influence adverse clinical outcomes. In practice, we identified three groups distinguished by their overall course, namely remained independent (RI), remained dependent (RD) and fransitioned from independent to dependent (DD). Therefore, in terms of dependency status, the most clinically interesting group was DD and the appropriate comparison group was RI, since both groups began independent and one transitioned to dependency whereas the other did not. The equivalent comparator for our RD group would have consisted of participants starting dependent and transitioning to independent but, perhaps unsurprisingly, no such group emerged. The RD group is then a group without an effective comparator, of interest primarily for clinical characterisation of the patient groups we distinguished.

We have, nevertheless, undertaken a repeat of our logistic regression analysis with the RD group included, which then generates a model of transition to dependency with as comparison group, all those who did not transition to dependency regardless of initial status. The great majority of the variables that had shown a significant association with DD in univariate analysis ceased to be significant at p<0.1. Only atrial fibrillation (p=0.06) and NIHSS (p=0.006) remained. On multivariable analysis only NIHSS remained signdificant at p=0.01, whether the analysis was undertaken with SD or with MAGE separately or together. The loss of some univariate significances is to be expected given that combining RI and RD creates a comparison group that includes participants from two very different patient groups, one relatively healthy and the other with more advanced clinical characteristics (table 2). To that extent, inclusion of RD patients will have blunted discrimination of significant associations by introducing inconsistent data for the initial condition with which transition was to be defined. For completeness, we have added a brief paragraph to the Results that acknowledges the characteristics of RD group (lines 197-204) but, overall, feel that giving the group any more prominence could be confusing.

Main Results

Page 11, Lines 193–198: The multivariate model's failure to show glycaemic variability as significant deserves deeper exploration. Were SD and MAGE highly correlated with NIHSS, thereby diminishing their impact?

Neither SD (r2 0.02) nor MAGE (r2 <0.0001) were correlated with NIHSS.

Discussion

Key Results

Page 13, Lines 200–208: The emphasis on NIHSS as the primary predictor of functional outcome is appropriate, but the failure of glycaemic variability metrics should be contextualized against prior conflicting findings.

I have added the following sentence:

‘Additionally, given the conflicting findings of previous studies, it is possible that GV and stroke outcomes are not intrinsically related and that, when identified, variable GV metrics are biomarkers for other pathophysiological processes or pharmacological interventions caused by or a result of having had a stroke. A larger study would help to discern this.’

Limitations

The limitation regarding single-center recruitment is mentioned but underemphasized. Additionally, the lack of data on specific interventions (e.g., use of intravenous glucose) during monitoring introduces potential bias.

Thank you; I have added the following sentences:

‘The focus for this hypothesis-generating study was on GV however it may accrue, thus although we did not collect data on the use of glucose containing intravenous fluids use or other interventions, all participants with diabetes had regular fingerprick glucose testing and management of blood glucose was undertaken according in line with local guidelines and clinical assessment by the treating medical team.’

And:

‘The study recruited from a single centre in which all participants had the benefit of consultant led multidisciplinary care in a dedicated hyperacute stroke unit, which may limit wider interpretation of the results to other stroke units or hospitals with different practices.’

Interpretation

The authors miss an opportunity to discuss the clinical implications of their findings. How might these results influence the use of CGM in stroke patients or guide future trias?

Thank you for raising this – a very fair point. I have added this to the discussion:

‘As the use of CGM becomes more widespread in people with type 2 diabetes, and in inpatient settings, challenges we encountered in undertaking this study highlight potentially important clinical implications.32,33 In particular, the continued use of CGM technology in patients requiring MRI imaging will require additional training and support of both inpatient and imaging staff. The CGM used in this study has been shown to be safe in MRI scanners although national guidelines currently recommend their removal.33,34 Looking ahead, until this tension is resolved, local guidance and education will be required to enable and implement minimally interrupted use of CGM in people who require MRI scans.’

32.   Garg SK, Hirsch IB, Repetto E, et al. Impact of continuous glucose monitoring on hospitalizations and glucose control in people with type 2 diabetes: real-world analysis. Diabetes Obes Metab 2024; 26(11): 5202-5210 doi:10.1111/dom.15866

33.   Avari P, Choudhary P, Lumb A, et al. Using technology to support diabetes care in hospital: Guidelines from the Joint British Diabetes Societies for Inpatient Care (JBDS-IP) group and Diabetes Technology Network (DTN) UK. Diabet Med 2024; 00:e15452 doi:10.1111/dme.15452

34.   Thomas C, Welsh JB, Lu S, et al. Safety and Functional Integrity of Continuous Glucose Monitoring Components After Simulated Radiologic Procedures. J Diabetes Sci Technol 2021;15(4):781-785 doi: 10.1177/1932296820920948

Supplementary Information

The inclusion of post hoc power calculations is commendable. However, presenting this information graphically could aid reader comprehension.

The post-hoc power calculations we present at the end of our Results section were really intended to illustrate the extent to which the numbers we recruited approached those needed for a fully powered study, rather than support for any recommendations as to how others should design their studies. The summary data we present and used for estimating numbers needed could be used by other investigators for powering their own studies, which could involve any one of many different combinations of anticipated study group size differences and anticipated group differences. In view of this, we are unsure as to how any graphic presentation of numbers needed might help. We could present numbers as a table but, again, numbers needed are so dependent on study design, we feel that even that would give the impression of over-determination as to how others should proceed.

Minor Issues

Formatting inconsistencies in tables (e.g., Table 2) detract from readability. Ensure consistent decimal places for all variables.

Done

Some abbreviations (e.g., MAG, TIR) are not defined upon first use in the main text.

These are defined in this paragraph in the Methods under Study Protocol:

Glucose data downloaded from receiver after sensor removal. Data within first 72 hours after admission were used to calculate mean glucose, percentage (%) time in range (TIR; 70–180 mg/dL), % time below range (TBR; <70 mg/dL and <54mg/dL), % time above range (TBR; >180 mg/dL) and GV including standard deviation (SD), % coefficient of variation (CV), mean absolute glucose (MAG), mean amplitude of glucose excursions (MAGE), continuous overlapping net glycaemic action (CONGA) and low blood glucose index (LBGI).

---

## [Decision Letter · Decision Letter 2]

16 Jan 2025

Glycaemic level and glycaemic variability in acute ischaemic stroke and functional outcome at discharge: an observational continuous glucose monitoring study

PONE-D-24-25866R2

Dear Dr. Oliver,

We’re pleased to inform you that your manuscript has been judged scientifically suitable for publication and will be formally accepted for publication once it meets all outstanding technical requirements.

Kind regards,

Atakan Orscelik

Academic Editor

PLOS ONE

Additional Editor Comments (optional):

Reviewers' comments:

Reviewer's Responses to Questions

**Comments to the Author**

1. If the authors have adequately addressed your comments raised in a previous round of review and you feel that this manuscript is now acceptable for publication, you may indicate that here to bypass the “Comments to the Author” section, enter your conflict of interest statement in the “Confidential to Editor” section, and submit your "Accept" recommendation.

Reviewer #3: All comments have been addressed

2. Is the manuscript technically sound, and do the data support the conclusions?

Reviewer #3: Yes

3. Has the statistical analysis been performed appropriately and rigorously? 

Reviewer #3: Yes

4. Have the authors made all data underlying the findings in their manuscript fully available?

Reviewer #3: Yes

5. Is the manuscript presented in an intelligible fashion and written in standard English?

Reviewer #3: Yes

6. Review Comments to the Author

Reviewer #3: (No Response)

7. PLOS authors have the option to publish the peer review history of their article (what does this mean? ). If published, this will include your full peer review and any attached files.

**Do you want your identity to be public for this peer review?** For information about this choice, including consent withdrawal, please see our Privacy Policy .

Reviewer #3: No

---

## [Editor Report · Acceptance letter]

PONE-D-24-25866R2

PLOS ONE

Dear Dr. Oliver,

I'm pleased to inform you that your manuscript has been deemed suitable for publication in PLOS ONE. Congratulations! Your manuscript is now being handed over to our production team.

Kind regards,

on behalf of

Dr. Atakan Orscelik

Academic Editor

PLOS ONE